# Somatic Tumor Profile Analysis in a Patient with Germline *PMS2* Mutation and Synchronous Ovarian and Uterine Carcinomas

**DOI:** 10.3390/jpm11070634

**Published:** 2021-07-05

**Authors:** Karen M. Huelsman, Jack B. Basil, Rebecca Sisson, Lindsay R. Lipe, Brett Mahon, David J. Draper

**Affiliations:** 1TriHealth Cancer Institute, Cincinnati, OH 45220, USA; Jack_basil@trihealth.com (J.B.B.); David_draper@trihealth.com (D.J.D.); 2Cincinnati Children’s Hospital Medical Center, Cincinnati, OH 45229, USA; Rebecca.Sisson@cchmc.org; 3Ambry Genetics, Aliso Viejo, CA 92656, USA; LLipe@ambrygen.com; 4Tempus, Chicago, IL 60654, USA; Brett.Mahon@tempus.com

**Keywords:** tumor profiling, Lynch syndrome, *PMS2* germline mutation, synchronous endometrial ovarian tumor, HRD, cancer genomics

## Abstract

Lynch syndrome patients with synchronous endometrial and ovarian cancer (SEOC) are rare. When these cases occur, they are most often endometrioid histology and early grade. Early-grade tumors are not often sent for somatic tumor profiling. We present a 39 year old SEOC patient with germline *PMS2* Lynch syndrome and clinical tumor analysis leading to insight regarding the origin and cause of these tumors, with potential therapy options. *PMS2*-related SEOC is less common due to lower risks for these cancers associated with germline *PMS2* mutation compared to other Lynch genes. While synchronous cancers are not common, they are more likely to occur with Lynch syndrome. Tumor profiling with next-generation sequencing of 648 genes identified sixteen shared somatic actionable and biologically relevant mutations. This case is a rare example of a patient with *PMS2* germline Lynch syndrome with shared somatic variants that demonstrate clonality of the two tumors arising from one common site.

## 1. Introduction

Cancer of the uterus is the fourth most common cancer diagnosed among females in the United States (US) when non-melanoma skin cancers are excluded [1]. Ovarian cancers (OC) are less common, but represent the fifth leading cause of cancer death in US females [1]. Lynch syndrome (LS) is caused by germline mutations in one of four mismatch repair genes (MMR) including *MLH1*, *MSH2*, *MSH6*, and *PMS2*, or in the epithelial cell adhesion molecule *EPCAM* which can cause LS by silencing *MSH2* [2,3]. While the majority of endometrial cancers (EC) are sporadic, about 4% are associated with germline Lynch mutations [2]. Patients with LS germline mutations are predisposed to several specific cancer types including colon, endometrial, ovarian, renal pelvis, bladder, gastric, and others. The NCCN Guidelines version 1.2021 state that the risk for each cancer varies with the specific germline mutation [3]. For example, patients with *MSH2* germline mutations have up to a 57% lifetime risk for endometrial and up to a 38% risk for ovarian cancer, with median age range from 43 to 48 years. However, patients with a *PMS2* mutation have lower cancer penetrance with up to a 26% lifetime risk for endometrial (median age 49–50 years) and up to a 3% lifetime risk (median age 51–59 years) for ovarian cancer [3,4].

Immunohistochemistry for protein expression of MMR is used to screen tumors for Lynch syndrome [5]. When a germline or somatic mutation in an MMR gene is present, tumors exhibit MMR deficiency, demonstrating the absence of corresponding MMR protein(s) [6,7]. Tumors with MMR deficiency also typically display high levels of microsatellite instability (MSI-high). Tumors with a solitary loss of PMS2 expression on IHC are associated with *PMS2* germline mutations [8,9].

Synchronous endometrial and ovarian cancer (SEOC) is rare, but has been reported in a case series with grade 1 endometrioid pathology subtype without molecular analysis [10]. A study of 32 SEOC patients concluded that most are sporadic and not caused by germline mutations, but authors did find three cases with MLH1 protein loss and one case with MSH6 MMR protein loss [11]. Case reports of SEOC in Lynch patients with *MSH2* germline mutations have been reported but are still rare, and most did not examine the full tumor profiling [12]. Moukarzel et al. performed massive parallel sequencing targeting 468 genes in four patients with germline Lynch mutations and concluded that sporadic SEOCs are more often clonally related, while those with LS may represent distinct primary tumors [13]. They did note that a subset of SEOCs may arise from a shared primary tumor, with the endometrium as the most likely origin. Somatic tumor profiling can provide molecular characterization and insight to tumorigenesis. We report a patient with germline *PMS2* mutation and SEOC with complete tumor profiling results. At initial workup, proficient MMR results on endometrial IHC staining were found that were not consistent with the germline *PMS2* Lynch mutation. Therefore, the clinical team ordered somatic tumor profiling to better understand this patient’s tumor and confirm Lynch association.

## 2. Case Presentation

A 39 year old female with a BMI (body mass index) of 57 presented to the ER with a history of back pain. Pelvic ultrasound identified a 5.7 cm right-sided complex mass and a normal size uterus. Abdominal–pelvic computerized tomography confirmed a 6.3 cm right-adnexal mass. Endometrial biopsy was performed, revealing a FIGO grade 1 endometrioid adenocarcinoma. Robotic hysterectomy with bilateral salpingo-oophorectomy, sentinel lymph node removal on the left, full lymphadenectomy on the right, and partial omentectomy was performed (JB). Final pathology revealed a 2.2 cm endometrioid adenocarcinoma, FIGO grade 2 without myometrial invasion, and a 4.0 cm mixed clear-cell/endometrioid left-ovarian adenocarcinoma. Zero of thirty-two lymph nodes were involved, and the omentum was negative. This patient had separate stage IA endometrial cancer (MMR proficient on IHC) and stage IA left ovarian cancer. The treatment team (DD, JB) attempted six cycles of adjuvant chemotherapy with carboplatin and Taxol; however, the patient experienced serious side effects and toxicity and after two separate attempts, opted for no further adjuvant therapy. Follow-ups with medical oncologist (DD) and gynecology oncologist (JB) have shown no signs of recurrent disease. The patient underwent her first colonoscopy with esophagogastroduodenoscopy (EGD). An ascending sessile serrated polyp and a transverse polyp described as benign polypoid colonic tissue with small lymphoid aggregates were identified. Stomach biopsy included benign gastric tissue with focal changes suggestive of early chemical/reactive gastropathy and minimal chronic inflammation, negative for evidence of *H. pylori* infection and negative for intestinal metaplasia, dysplasia, and malignancy. Germline and somatic tumor profiling was performed according to the methods section.

## 3. Methods

The CancerNext^®^ +RNAinsight^®^ germline test at Ambry Genetics lab provided an analysis of 34 genes associated with hereditary cancer predisposition. Genomic deoxyribonucleic acid (gDNA) and ribonucleic acid (RNA) were isolated from the patient specimen using standardized methodology and quantified. RNA was converted to complementary DNA (cDNA) by reverse-transcriptase polymerase chain reaction (RT-PCR). Sequence enrichment of the targeted coding exons and adjacent intronic nucleotides was carried out via a bait-capture methodology using long biotinylated oligonucleotide probes followed by polymerase chain reaction (PCR) and next-generation sequencing. Additional DNA analyses included Sanger sequencing for any regions missing or with insufficient read depth coverage for reliable heterozygous variant detection. Variants in regions complicated by pseudogene interference, variant calls not satisfying depth of coverage and variant allele frequency quality thresholds, and potentially homozygous variants were verified by Sanger sequencing. The *BRCA2* Portuguese founder mutation, c.156_157insAlu (also known as 384insAlu), and the *MSH2* coding exons 1–7 inversion were detected by next-generation sequencing and confirmed by multiplex ligation-dependent probe amplification (MLPA) or PCR and agarose gel electrophoresis. Gross deletion/duplication analysis for 30 of the genes (excluding *HOXB13*, *PMS2*, *POLD1*, and *POLE*) was performed using a custom pipeline based on read depth from NGS data and/or targeted chromosomal microarray with confirmatory MLPA when applicable. Gross deletion/duplication analysis of *PMS2* was performed using MLPA kit P008-B1. If a deletion was detected in exons 13, 14, or 15 of *PMS2*, double-stranded sequencing of the appropriate exon(s) of the pseudogene *PMS2CL* was performed to determine whether the deletion was located in the *PMS2* gene or pseudogene. All sequence analysis was based on the following NCBI reference sequences: *APC*- NM_000038.5 & NM_001127511.2, *ATM*- NM_000051.3, *BARD1*- NM_000465.2, *BMPR1A*- NM_004329.2, *BRCA1*- NM_007294.3, *BRCA2*- NM_000059.3, *BRIP1*- NM_032043.2, *CDH1*- NM_004360.3, *CDK4*- NM_000075.3, *CDKN2A*- NM_000077.4 and NM_058195.3 (p14ARF), *CHEK2*- NM_007194.3, *DICER1*- NM_177438.2, *HOXB13*- NM_006361.5, *MUTYH*- NM_001128425.1, *MRE11A* NM_005591.3, *MLH1*- NM_000249.3, *MSH2*- NM_000251.1, *MSH6*- NM_000179.2, *NBN*- NM_002485.4, *NF1*- NM_000267.3, *PALB2*- NM_024675.3, *PMS2*- NM_000535.5, *POLD1*- NM_002691.2, *POLE*- NM_006231.2, *PTEN*- NM_000314.4, *RAD50*- NM_005732.3, *RAD51C* NM_058216.1, *RAD51D*- NM_002878.3, *SMAD4*- NM_005359.5, *SMARCA4*- NM_001128849.1, *STK11*- NM_000455.4, *TP53*- NM_000546.4.

Analytical range: The CancerNext^®^ +RNAinsight^®^ test was used to target detection of DNA sequence mutations in 32 genes (*APC*, *ATM*, *BARD1*, *BMPR1A*, *BRCA1*, *BRCA2*, *BRIP1*, *CDH1*, *CDK4*, *CDKN2A*, *CHEK2*, *DICER1*, *HOXB13*, *MLH1*, *MSH2*, *MSH6*, *MUTYH*, *MRE11A*, *NBN*, *NF1*, *PALB2*, *POLD1*, *POLE*, *PMS2*, *PTEN*, *RAD50*, *RAD51C*, *RAD51D*, *SMAD4*, *SMARCA4*, *STK11*, and *TP53*) by either next-generation or Sanger sequencing of all coding domains and well into the flanking 5′ and 3′ ends of all the introns and untranslated regions. For *HOXB13*, only variants impacting codon 84 are routinely reported. For *POLD1* and *POLE*, only missense and in-frame indel variants in the exonuclease domains (codons 311–541 and 269–485, respectively) are routinely reported. Gross deletion/duplication analysis determines gene copy number for the covered exons and untranslated regions of sequenced genes (excluding *HOXB13*, *POLD1*, and *POLE*) as well as *GREM1* and *EPCAM*. For *GREM1*, only the status of the 40 kb 5′UTR gross duplication is analyzed and reported. For *EPCAM*, only gross deletions encompassing the 3′ end of the gene are reported. For *APC*, all promoter 1B gross deletions as well as single-nucleotide substitutions within the promoter 1B YY1 binding motif (NM_001127511 c.-196_-186) are analyzed and reported. RNA transcripts were screened for 18 genes (*APC*, *ATM*, *BRCA1*, *BRCA2*, *BRIP1*, *CDH1*, *CHEK2*, *MLH1*, *MSH2*, *MSH6*, *MUTYH*, *NF1*, *PALB2*, *PMS2* exons 1–10, *PTEN*, *RAD51C*, *RAD51D*, and *TP53*) and compared to a human reference pool. The absence or presence of RNA transcripts meeting quality thresholds were incorporated as evidence towards assessment and classification of DNA variants. Any regions not meeting RNA quality thresholds were excluded from analysis. Regions routinely excluded due to chronically low expression in human peripheral lymphocytes include: *BRCA2* (exon 1), *BRIP1* (exons 18, 20), *CDH1* (exons 1, 2, 16), and *CHEK2* (exons 1, 7, 8).

Tempus Lab PD-L1 tumor staining was performed on the endometrial and ovarian tissue samples. PD-L1 is defined as complete circumferential and/or partial linear plasma membrane staining of tumor cells at any intensity. Tumor-associated immune cell staining was defined as membrane and/or cytoplasmic staining (at any intensity) of mononuclear inflammatory cells (MICs) within tumor nests and adjacent supporting stroma. PD-L1 IHC 22C3 pharmDx is a qualitative immunohistochemical assay using Monoclonal Mouse Anti-PD-L1, Clone 22C3 intended for use in the detection of PD-L1 protein in formalin-fixed, paraffin-embedded (FFPE) non-small-cell lung cancer (NSCLC), gastric or gastroesophageal junction (GEJ) adenocarcinoma, esophageal squamous cell carcinoma, cervical cancer, and urothelial carcinoma tissues. Scoring is not provided in tumors for which no scoring system has been published. See the KEYTRUDA^®^ product label for expression cutoff values guiding therapy in specific clinical circumstances. The same DAKO PD-L1 22C3 clone was used for both tumors. 

The initial IHC for MMR protein expression test was run at TriHealth pathology lab on an endometrial sample from a total abdominal hysterectomy with bilateral salpingo-oophorectomy. This demonstrated intact nuclear expression for MLH1, MSH2, MSH6, and PMS2. Background nonneoplastic tissue/internal control was run with intact nuclear expression. A second IHC for MMR protein expression was run at Tempus Labs using the same endometrial tissue block. 

Somatic tumor profiling was performed by Tempus Labs on endometrial and ovarian tissue samples, with an accompanying matched normal blood sample. The Tempus|xT next-generation sequencing assay is a CAP/CLIA-validated panel designed to detect actionable oncological targets by sequencing FFPE tumor samples with matched normal saliva or blood samples, when available. The Tempus|xT assay includes DNA sequencing of 648 genes spanning ~3.6 Mb of genomic space and full-transcriptome RNA sequencing. From DNA sequencing, somatic and incidentally detected germline single-nucleotide variants (SNVs), insertions and deletions (indels), and copy number variants (CNVs) are detected. Additionally, translocations in 22 genes are detected, along with two promoter regions (*PMS2* and *TERT*) and 239 sites used to determine microsatellite instability (MSI) status. Tumor mutational burden (TMB) is calculated as described below. Some viral sequences, such as HPV and EBV, may be reported to offer a diagnostic or prognostic insight when deemed appropriate by Tempus pathologists. From RNA-seq, gene fusions (translocations) are detected in an unbiased and comprehensive manner. Full-transcriptome RNA expression counts are analytically validated. The Tempus|xT assay requires specimens with a tumor content of 20% post macrodissection (minimum 30% for MSI status). Clinical sequencing is performed to 500× depth of coverage for tumor specimens and 150× for normal specimens. Performance specifications and a complete gene list are available online at https://www.tempus.com/genomic-profiling/#proprietary-sequencing (accessed on 15 April 2021).

The Tempus|xT assay identifies variants by aligning the patient’s DNA sequence to the human genome reference sequence version hg19 (GRCh37) and classifies each variant as potentially actionable, biologically relevant, variant of unknown significance (VUS), or benign. Variants considered potentially actionable alterations are protein-altering variants with an associated therapy based on evidence from the medical literature. Biologically relevant alterations are protein-altering variants that may have functional significance or have been observed in the medical literature, but are not associated with a specific therapy in the Tempus knowledge database. VUSs are protein-altering variants exhibiting an unclear effect on function and/or without sufficient evidence to determine their pathogenicity. Benign variants are not reported. The clinical summary shows actionable and biologically relevant somatic variants, and certain pathogenic or likely pathogenic inherited variants that are reported as incidental findings when applicable. Reportable secondary/incidental findings are limited to genes and variants associated with inherited cancer syndromes. Germline genes that are reported include: *APC*, *ATM*, *AXIN2*, *BMPR1A*, *BRCA1*, *BRCA2*, *BRIP1*, *CDH1*, *CDKN2A*, *CEBPA*, *CHEK2*, *EGFR*, *EPCAM*, *ETV6*, *FH*, *FLCN*, *GATA2*, *MEN1*, *MLH1*, *MSH2*, *MSH3*, *MSH6*, *MUTYH*, *NBN*, *NF2*, *PALB2*, *PMS2*, *POLD1*, *POLE*, *PTEN*, *RAD51C*, *RAD51D*, *RB1*, *RET*, *RUNX1*, *SDHAF2*, *SDHB*, *SDHC*, *SDHD*, *SMAD4*, *STK11*, *TP53*, *TSC1*, *TSC2*, *VHL*, and *WT1*.

TMB calculated by the Tempus|xT assay measures the quantity of somatic mutations of any pathogenicity, including benign, carried in a tumor as the number of single-nucleotide protein-altering mutations per million coding base pairs. TMB is calculated at the time of initial report delivery. Accordingly, the TMB calculation is based upon (a) both the tumor and normal sample if Tempus had analyzed both at the time of the initial report, or (b) the tumor sample only if no normal sample had been analyzed at the time of the initial report. MSI refers to hypermutability caused by genetic or acquired defects in the DNA mismatch repair pathway. MSI status is divided into MSI-high (MSI-H), microsatellite stable (MSS), and microsatellite equivocal (MSE) tumors. MSI-H tumors have changes in microsatellite repeat lengths due to defective DNA mismatch repair activity, MSS tumors do not have detectable defects in DNA mismatch repair, and MSE tumors have an intermediate phenotype which cannot be clearly classified as MSI-H or MSS based on the statistical cutoff used to define those categories. If MSI status will affect clinical management, immunohistochemical staining for DNA mismatch repair proteins, or application of another method for ascertaining MSI status, is recommended.

Homologous recombination deficiency (HRD) status was determined for the endometrial and ovarian tissue samples using the Tempus|HRD assay. The Tempus|HRD assay computational algorithm uses results from tumor and normal matched xT sequencing data to calculate the genome-wide loss of heterozygosity (GWLOH) percentage and uses the somatic and germline alteration status of *BRCA1* and *BRCA2* to determine HRD status. GWLOH is calculated by determining the percentage of genomic segments with LOH by the Tempus copy number calling algorithm (CONA). GWLOH is considered positive for HRD at ≥29% for breast cancer, ≥25% for ovarian cancer, ≥28% for pancreatic cancer, and ≥33% for any other cancer type (endometrial). *BRCA1* and *BRCA2* alterations considered positive for HRD include the following: a pathogenic or likely pathogenic alteration with LOH, biallelic pathogenic or likely pathogenic alterations, or two-copy loss. *BRCA1/2* LOH is computed using CONA, which uses tumor purity and copy states in the tumor genome to generate copy number status. HR-pathway genes analyzed on the Tempus|xT panel include: *ATM*, *BARD1*, *BRCA1*, *BRCA2*, *BRIP1*, *CDK12*, *CHEK1*, *CHEK2*, *FANCA*, *FANCL*, *HDAC2*, *MRE11*, *NBN*, *PALB2*, *RAD51B*, *RAD51C*, *RAD51D*, and *RAD54L*. 

## 4. Results

Family history included three cases of breast cancer in early 50s, pancreatic, and a gastrointestinal cancer. There were no known cases of colon cancer or other gynecological cancers (Figure 1). Patient met with a genetic counselor (RS) and consented to germline genetic testing via a 34 gene germline cancer panel at Ambry Genetics lab. Testing identified a germline likely pathogenic *PMS2* mutation c.2095G>C p.D699H mutation and a *PALB2* VUS c.1250C>A p.S417Y. 

Patient underwent germline testing at Ambry Genetics lab with a 34 gene CancerNext +RNAinsight^®^ panel, identifying a likely pathogenic *PMS2* mutation and a *PALB2* VUS. This *PMS2* mutation is classified as likely pathogenic on ClinVar. The location of the *PMS2* germline mutation c.2095G>C p.D699H mutation was at the 3′ end of the *PMS2* gene in exon 12. The *PMS2* mutation was classified as likely pathogenic in ClinVar database. This is an area with high homology to the *PMS2CL* pseudogene [14]. The Tempus next-generation sequencing (NGS) assay is unable to distinguish with certainty *PMS2* from *PMS2CL*, so the *PMS2* pathogenic result was not reported on the limited set of germline genes reported by Tempus. To distinguish the pseudogene, Ambry Genetics used a combination of long-range PCR and a multiplex ligation-dependent PCR amplification (MLPA). The isoforms used by each laboratory are available by request.

The germline *PALB2* variant was classified as a VUS based on the following information from Ambry lab: The p.S417Y variant (also known as c.1250C>A), located in coding exon 4 of the *PALB2* gene, results from a C to A substitution at nucleotide position 1250. The serine at codon 417 is replaced by tyrosine, an amino acid with dissimilar properties. Functional studies have demonstrated that S417Y leads to a partial reduction of ChAM-mediated PALB2 chromatin association without affecting the cellular resistance to CPT [15]. This amino acid position is highly conserved in available vertebrate species. In addition, the in silico prediction for this alteration was inconclusive. Since supporting evidence is limited at this time, the clinical significance of this alteration remains unclear.

Because the proficient MMR result on endometrial IHC for MMR expression was not consistent with the germline PMS2 Lynch mutation, the clinical team (KH, DD) offered somatic tumor profiling to better understand this patient’s tumors and to determine whether this tumor was indeed Lynch-associated. The patient consented to Tempus|xT solid tumor profiling with a matched tumor normal blood sample. The tumor sample from the left fallopian tube and ovary of mixed endometrioid adenocarcinoma and clear-cell carcinoma and the uterine specimen of endometrioid adenocarcinoma were both sent to Tempus for xT tumor profiling. Tempus|xT targeted somatic tumor profiling of 648 genes was performed. This included IHC for PD-L1 expression (Figure 2) and MMR (Figure 3), TMB, homologous recombination deficiency (HRD), somatic, and germline analysis. 

TMB was 245.8 mut/MB in ovarian and 330.5 mut/MB in endometrial with high microsatellite instability status (MSI-H) for both tumors. There was no recognized second hit, or mutation, identified in *PMS2*. This very high TMB is considered an ultrahypermutator tumor by Campbell et al. [16]. It was noted that a *POLE* variant was identified on tumor profile testing in both tumors (*POLE* c.1306C>T p.P436S) and that polymerase proofreading alterations in *POLE* can play a role in ultrahypermutator phenotypes. The *POLE* c.1306C>T p.P436S variant was reviewed by a Tempus Lab variant scientist and functional evidence led to a somatic pathogenic classification. The evidence used to classify *POLE* c.1306C>T p.P436S as pathogenic included: 

1. Variant causes increased mutagenesis in yeast assays [17]; variant is referred to as p.P451S in yeast).

2. Variant identified in a patient whose phenotype is similar to patients with known *POLE* mutations [18]. 

3. Variant falls in the *POLE* exonuclease domain (codons 269–485). While this evidence is sufficient to classify the variant as pathogenic in a somatic context, it does not quite meet the threshold for being classified pathogenic as a germline variant and is seen as germline VUS in ClinVar. We cannot be certain that the ultrahypermutator phenotype was caused by *PMS2*, *POLE*, or a synergistic effect. The fact that the patient had such a high TMB could be consistent with this *POLE* variant driving the hypermutator phenotype. Additionally, the *POLE* mutation is associated with high proportions of C>A, C>T, and T>G variants [19], and a large number of these were observed in this patient’s case. Alexandov has described mutational signatures caused by mutations in the exonuclease domain of *POLE*, including single-base substitutions SBS10a, SBS10b, and SBS28, referenced in the *Catalogue of Somatic Mutations in Cancer*. Tumors with these mutational profiles generate a large volume of somatic mutations (>100 mut/MB) and are termed hypermutators [20,21,22].

*MLH1* promoter hypermethylation is not a standard component of Tempus xT testing. *BRAFV600E* was included in the somatic tumor profiling and was not identified. IHC assessment of MMR gene protein expression was also performed and was reported as normal on both tumors. The immunohistochemical staining performed at Tempus Labs noted that *PMS2* expression was weak in the uterine tumor, but was still considered normal (Figure 3).

Both tumors had a very large number of somatic variants. The ovarian tumor had 5 potentially actionable and 21 biologically relevant variants. The uterine tumor had 7 potentially actionable and 27 biologically relevant variants (Figure 4). In addition, the ovarian tumor had 506 VUSs and the endometrial tumor had 657 VUSs. When variants were compared between tumors, there were many variants shared between them. For example, there were three potentially actionable somatic variants in common between the ovarian and uterine tumors: *BRCA1* p.W312*, *ATM* p.I1270fs, and *PIK3CA* p.R93Q. The *POLE* p.P436S variant was also seen in both tumors but was classified as potentially actionable in the uterine cancer and as a VUS in the ovarian cancer. There were also 13 shared variants among the biologically relevant variants (Figure 4). The ovarian tumor had 26 potentially actionable and biologically relevant variants, of which 16 (61.5%) were in common with the endometrial tumor. The endometrial tumor had 34 potentially actionable and biologically relevant variants, of which 16 (47%) were in common with the ovarian tumor. It was confirmed by Ambry Genetics that none of these other common gene variants, among the 34 genes tested at Ambry, were germline. 

Homologous recombination deficiency (HRD) was also evaluated via genome-wide loss of heterozygosity (LOH) in each tumor (Figure 5). While both tumors demonstrated pathogenic/likely pathogenic genomic variants in the HRD pathway (*BRCA1/2* and *ATM*), the genome-wide loss of heterozygosity (LOH) was well below the threshold of 33% and was deemed “not detected”.

The tumor profile report gave multiple therapy options for this patient, including pembrolizumab, which is FDA-approved for the current diagnosis based on MSI-H and TMB, PARP inhibitor based upon somatic *BRCA1*, *BRCA2*, and *ATM* variants, and alpelisib based on the *PIK3CA* variant. In addition, the uterine tumor report had FDA-approved therapies for other indications: everolimus and sirolimus for mTOR inhibitors and nivolumab based on the *POLE* variant. 

## 5. Discussion

This patient had a personal and family history of cancer that was not consistent with the moderate-risk *PMS2* Lynch syndrome risk profile in NCCN [3]. The Prospective Lynch Syndrome Database has reported primarily colorectal and endometrial cancer risk to be associated with *PMS2* [23]. This patient’s personal and family history included ovarian cancer, three cases of breast cancer, and pancreatic cancer. These are all at lower risk in *PMS2* germline carriers, but breast cancers have been reported in some *PMS2* Lynch patients [24]. Although *PMS2* mutation carriers do have lower overall cancer risk compared to other Lynch mutations carriers, a smaller study of *PMS2* LS patients showed that 60% of patients with MMR-deficient/MSI tumors presented with extracolonic cancers [25]. The NCCN guidelines primarily address surveillance and prevention strategies for colon and gynecological cancers with *PMS2* germline mutation. Clinicians may consider family history to determine any indication for screening other cancers.

The location of the *PMS2* mutation c.2095G>C p.D699H mutation was in an area with high homology to the *PMS2CL* pseudogene. The tumor profiling lab was unable to distinguish the germline *PMS2* from pseudogene, demonstrating the importance of high-quality germline analysis beyond sequencing. In this case, both the germline and the somatic lab provided crucial information. Ambry Genetics lab reported that in vitro studies of this 3′ mutation showed expression levels similar to wildtype with reduced protein function. This likely explains why PMS2 protein expression was proficient on IHC in this patient’s tumor, albeit weakly. Despite the PMS2 expression on IHC, the tumors both had high TMB and MSI-H, confirming that these tumors were both Lynch-associated. 

The high TMB in both tumors was categorized as ultrahypermutator [16]. It was noted that the TMB in the endometrial tumor was higher at 330.5 mut/MB than the ovarian at 245.8 mut/MB, and that a *POLE* variant was present in both tumors. This very high TMB may have been caused by the germline *PMS2* mutation, and the somatic *POLE* polymerase proofreading alteration may also have played a role in tumorigenesis. 

The ability to compare somatic mutations shared by the endometrial and the ovarian tumors gave insight to a shared origin. The ovarian tumor had 16/26 or 61.5% potentially actionable and biologically relevant variants in common with the endometrial tumor, and the endometrial tumor had 16/34 or 47% of potentially actionable and biologically relevant variants in common with the ovarian tumor. Both tumors shared three potentially actionable variants in *ATM*, *BRCA1* and *PIK3CA*, as well as 13 biologically relevant variants in *APC*, *ATR*, *CREBBP*, *CTCF*, *HNF1A*, *KEAP1*, *KMT2D*, *NCOR1*, *NF1*, *PTEN*, *PTPN13*, *TBX3*, and *TP53* (Figure 4). In addition to the potentially actionable and biologically relevant variants, the number of VUSs was also reported for each tumor. The ovarian tumor had 506 VUSs and the endometrial tumor had 657 VUSs. We were unable to compare the VUSs directly for commonality.

Deshpande et al. demonstrated that better understanding of the pathways leading to MSI-high gynecological cancers will improve prediction of cancer progression and therapeutic response [26]. Takeda et al. stated that most clinically diagnosed cases of SEOC have clonally related cancers, indicating metastatic cancer [12]. Niskakoski et al. performed deep sequencing of 578 genes in five synchronous Lynch carcinomas with germline *MLH1* and *MSH2* [27]. The group found that synchronous cancers were concordant molecularly, suggesting shared origins in SEOC in Lynch syndrome. Moukarzel evaluated a series of five patients with germline MMR, looking at the clonal relationship and directionality of progression in SEOC and comparing to patients with sporadic SEOC [13]. They concluded that the directionality of progression was likely from the endometrium to the ovary. They found evidence that SEOC in LS patients may represent distinct primary tumors, which is logical considering the genetic predisposition. However, Moukarzel et al. did conclude that a subset of LS-associated SEOCs may originate from a single primary tumor, with endometrium being the most likely origin. Our case is a rare example of SEOC with germline *PMS2* mutation with 16 shared actionable and biologically relevant somatic variants that demonstrate clonality of the two carcinomas arising from one common site, most likely the endometrium. 

## Figures and Tables

**Figure 1 jpm-11-00634-f001:**
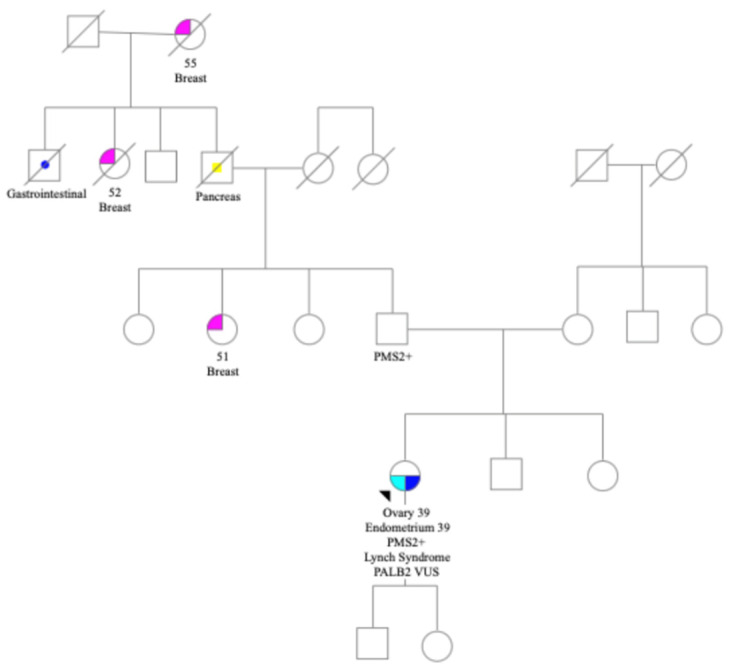
Patient pedigree illustrating family history of cancer.

**Figure 2 jpm-11-00634-f002:**
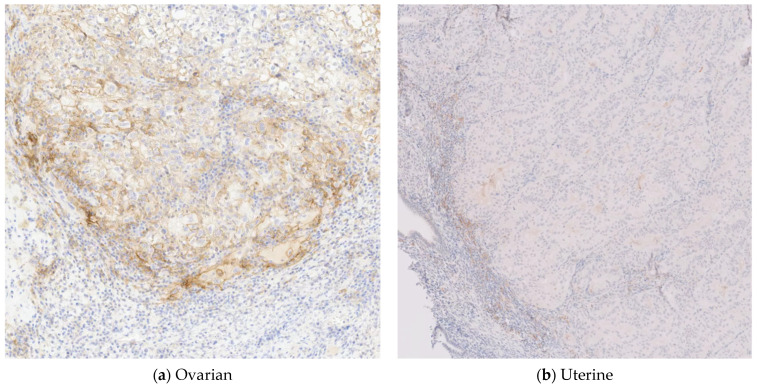
PD-L1 (22C3) expression of the (**a**) ovarian tumor showed 10% tumor cell positivity, while the (**b**) uterine tumor was negative.

**Figure 3 jpm-11-00634-f003:**
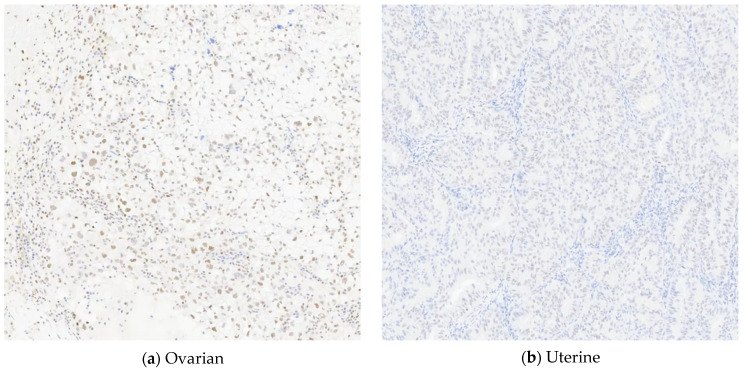
PMS2 (A16-4) immunohistochemical staining was intact for the (**a**) ovarian tumor and weak for the (**b**) uterine tumor.

**Figure 4 jpm-11-00634-f004:**
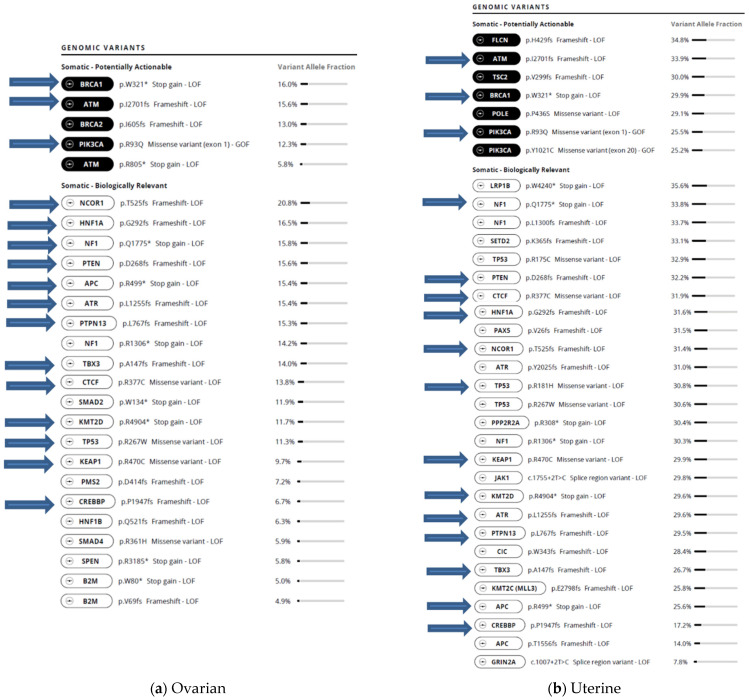
Somatic potentially actionable variants and biologically relevant variants (common variants shown with blue arrows) with variant allele fractions in (**a**) ovarian and (**b**) uterine tumors. A nonsense variant that removes the entire C-terminal part of the protein at the site of the variant is denoted with a * after the relevant amino acid.

**Figure 5 jpm-11-00634-f005:**
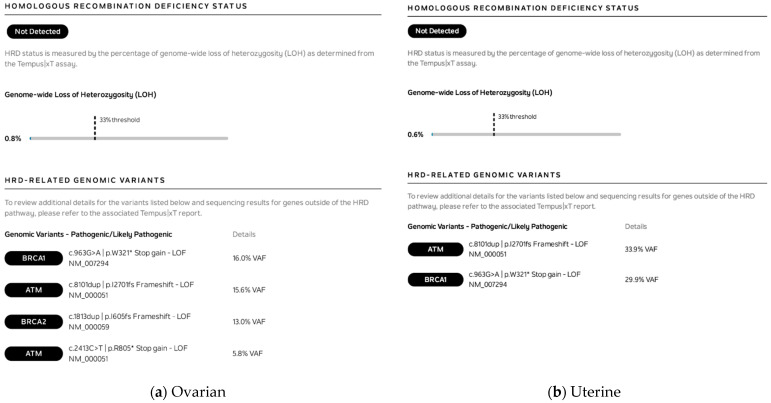
Homologous recombination deficiency LOH for (**a**) ovarian and (**b**) uterine tumors. (VUSs not shown.) A nonsense variant that removes the entire C-terminal part of the protein at the site of the variant is denoted with a * after the relevant amino acid.

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
