# Peer review of "Somatic Tumor Profile Analysis in a Patient with Germline PMS2 Mutation and Synchronous Ovarian and Uterine Carcinomas"

_jpm, 2021, doi:10.3390/jpm11070634_

Round 1
Reviewer 1 Report
In this report, the authors describe an interesting clinical case of a patient with synchronous endometrial and ovarian cancer. They performed genetic testing for this patient, both in germline and somatic DNA, providing additional insights about potential biologically relevant and actionable mutations, that could be potentially benefitting treatment decision-making.
I found very interesting the use of somatic mutational profiling as a tool to better understand the inconsistency between the PMS2 germline mutation found and the lack of MMR deficiency according to the IHC results, as well as the recommendation to consider screening for extracolonic cancers in Lynch syndrome mutations carriers. However, in my opinion, the authors should improve the description of the methods used for the study, especially regarding the next generation sequencing panels performed for a better assessment of the results obtained.
Major comments
- The methods section is completely missing for the genetic analysis, apart from a comprehensive description of the medical procedures and treatments. This would be very important to assess the reliability of the results in both germline and somatic next generation sequencing gene panels performed.
- Not sure about what the authors mean with “Mismatch Repair (MMR) was also performed” in line 121. Did they perform IHC, assessment of germline and somatic variants in all MMR genes, or any other analysis?
- Not sure about what authors consider potentially actionable or biologically relevant genetic variants. As in previous points, a clarification of the methodology is strongly recommended.
- The presence or absence of a somatic second hit for PMS2 in both tumors is not directly mentioned in the manuscript despite being the first goal of the somatic profiling performed. According to the results displayed it seems that no second hit was found, therefore confirming the IHC results and that the tumor may not be related to this PMS2 germline mutation. Could it be possible that the MSI-high is related to genetic/epigenetic somatic alterations in other MMR genes? Have the authors checked hypermethylation of the MLH1 promoter?
- If the TMB reported is correct, which is difficult to evaluate since the methods used for its calculation were not described, this means that both tumors fall in the category of ultrahypermutator tumors (according to Campbell et al. 2017 Cell, https://doi.org/10.1016/j.cell.2017.09.048), commonly caused by polymerase proofreading alterations in POLE/POLD1. This was completely ignored by the authors in their results and discussion sections, even though they reported a somatic mutation in the exonuclease domain of the polymerase epsilon gene (POLE) (which is a variant of unknown significance according to ClinVar: https://www.ncbi.nlm.nih.gov/clinvar/variation/575782/, also not reported by the authors in their manuscript). Interestingly, since the results indicate that the tumor may not be related to the PMS2 germline mutation, another hypothesis could be indicated for explaining the high number of mutations observed. Also, the mention of the TMB percentile is not appropriate, not sure what percentile the authors are referring to.
- The mention of the number of pages (supposedly of the genetic testing report?) is unnecessary and it would be preferable to give the exact number of somatic variants identified in both tumors for all categories indicated in the manuscript (instead of the broad statement “Both tumors had a very large number of somatic variants”). In contrast to what the authors mention (“The variants of uncertain significance were too numerous to count”), it is very important to count all somatic variants present in a tumor, in order to give an accurate number for the TMB, as well as to perform further genomic analysis. In fact, additional analyses (e.g., second hit testing, mutational signatures analysis) could be performed by using the germline-somatic next generation sequencing data generated to verify some of the authors’ hypothesis (maybe following some recommendation in Walsh et al. 2018 https://doi.org/10.1002/humu.23640). However, they would be out of the scope of the current case report.
- An appropriate comparison between the somatic mutations found in both tumors would also be helpful to clarify the similarities and differences between them, as well as to more clearly infer the alteration that came first. In this regard, the authors claimed in the last sentence of the manuscript that both carcinomas arose from one common site (endometrium) without further discussion or explanation. The authors merely indicated that there are 13 shared variants among one of the classes reported, but the lack of mention about percentages, other variant classes or variant allele frequencies, makes it difficult to assess the relevance of this finding.
Minor comments
- There are many issues regarding format, including for example figure legends showed before the figure (and not after it), errors in the references (ref. 3 - “In:”, ref. 10 & 13 – inconsistent doi, ref. 14 – inconsistent page number, etc.), abbreviations explained more than once (e.g., MMR), extra spaces in some sentences (e.g., line 160), numbers for the two tumors without clarification of which one is corresponding to each (e.g., TMB – line 120) an inconsistent reference within the text (ref. 2 – line 32) and missing methods section as mentioned above. The authors should carefully revise the format of their manuscript before resubmission.
- The authors should avoid the terms “full/comprehensive tumor analysis/profiling” since they performed gene panel sequencing analysis of both tumors of the patient, which covered less than 1,000 genes and therefore less than 1% of the whole genome of the tumors. Although they are implementing in the clinic a technique that substantially increases the number of genes tested in comparison with previous methods, a comprehensive tumor characterization would require a whole genome sequencing analysis.
- Germline and somatic mutations identified should be compared/reviewed/assessed using publicly available databases (OMIM, ClinVar, gnomAD, COSMIC, etc.) apart from the ones used by the genetic testing companies performing the analyses.
- The third paragraph of the results section should come before the second paragraph since it is about the germline testing mentioned in the first paragraph of the section.
- There are no mentions of figures 2 or 3 in the text of the manuscript.
- Reference 1 should be changed for a more up-to-date study.
- References are missing for the ovarian cancer epidemiology (line 27) and Lynch syndrome (line 28) statements in the introduction.
- Reference for Ambry Genetics statement in lines 170 and 171 is missing.
- CT and EGD abbreviations should be explained before use.
- The resolution of all figures should be improved.
- Typos: *Immunohistorychemistry (line 37), *this (line 159). Inconsistency: extra-colonic – extracolonic (line 164).
Author Response
REVIEWER 1 Reply from Authors
Thank you for your time and support in reviewing.
1 The methods section is completely missing for the genetic analysis, apart from a comprehensive description of the medical procedures and treatments. This would be very important to assess the reliability of the results in both germline and somatic next generation sequencing gene panels performed.
1Reply: Thank you for your review and feedback. We have added significant methodology in the following areas so that results can be better assessed:
- Full methodology for germline testing at Ambry lab
- Full methodology for Tempus somatic lab
- IHC for MMR protein expression
- TMB and MSI
- HRD
2 Not sure about what the authors mean with “Mismatch Repair (MMR) was also performed” in line 121. Did they perform IHC, assessment of germline and somatic variants in all MMR genes, or any other analysis?
2Reply: Yes, will change wording to “IHC assessment of MMR gene protein expression.”
3 Not sure about what authors consider potentially actionable or biologically relevant genetic variants. As in previous points, a clarification of the methodology is strongly recommended.
3 Reply: Thank you. This was added to methodology. Tempus lab classifies variants such that Potentially Actionable alterations are protein-altering variants with an associated therapy based on evidence from the medical literature. Biologically Relevant alterations are protein-altering variants that may have functional significance or have been observed in the medical literature but are not associated with a specific therapy in the Tempus knowledge database. Variants of Unknown Significance (VUSs) are protein-altering variants exhibiting an unclear effect on function and/or without sufficient evidence to determine their pathogenicity. Benign variants are not reported.
4 The presence or absence of a somatic second hit for PMS2 in both tumors is not directly mentioned in the manuscript despite being the first goal of the somatic profiling performed. According to the results displayed it seems that no second hit was found, therefore confirming the IHC results and that the tumor may not be related to this PMS2 germline mutation. Could it be possible that the MSI-high is related to genetic/epigenetic somatic alterations in other MMR genes? Have the authors checked hypermethylation of the MLH1 promoter?
4 Reply: We agree- no second hit was identified.
It is possible that the MSI high could be related to other factors.
MLH1 promoter hypermethylation is not a standard component of this test. BRAF V600 was included in the somatic tumor profiling and was not identified. BRAFV600E accounts for about 70% of tumors with hypermethylation of the MLH1 promoter. Our clinical practice is to run the MLH1 hypermethylation test only when IHC show loss of MLH1/PMS2.
Our hypothesis is that the MSI-high is associated with the germline PMS2 mutation but the IHC for MMR protein expression is normal (or near normal) because the PMS2 mutation is very 3’ and may not affect the protein as significantly as other mutations.
5 If the TMB reported is correct, which is difficult to evaluate since the methods used for its calculation were not described, this means that both tumors fall in the category of ultrahypermutator tumors (according to Campbell et al. 2017 Cell, https://doi.org/10.1016/j.cell.2017.09.048), commonly caused by polymerase proofreading alterations in POLE/POLD1. This was completely ignored by the authors in their results and discussion sections, even though they reported a somatic mutation in the exonuclease domain of the polymerase epsilon gene (POLE) (which is a variant of unknown significance according to ClinVar: https://www.ncbi.nlm.nih.gov/clinvar/variation/575782/, also not reported by the authors in their manuscript). Interestingly, since the results indicate that the tumor may not be related to the PMS2 germline mutation, another hypothesis could be indicated for explaining the high number of mutations observed. Also, the mention of the TMB percentile is not appropriate, not sure what percentile the authors are referring to.
5 Reply: Thank you, we have added additional methodology for the TMB to allow readers to further assess. The labels of the TMB scores were also adjusted to be clearer (mut/MB). Thank you for pointing out the idea of ultrahypermutator tumors. At 246 and 330 mutations/MBase, both tumors would be considered ultrahypermutator tumors, according to the Campbell article. This was added to result and discussion.
We also appreciate that you pointed out the POLE variant as another cause of mutator phenotype that we hadnot described. The POLE c.1306C>T p.P436S variant was reviewed by Tempus Lab variant scientist and functional evidence led to a somatic pathogenic classification. The evidence used to classify POLE c.1306C>T p.P436S as somatic pathogenic includes:
1. Variant causes increased mutagenesis in yeast assays [16]; variant is referred to as p.P451S in yeast).
2. Variant identified in a patient whose phenotype is similar to patients with known POLE mutations[17].
3. Variant falls in the POLE exonuclease domain (codons 269-485). While this evidence is sufficient to classify the variant as pathogenic in a somatic context, it does not quite meet the threshold for being classified pathogenic as a germline variant and is seen as germline VUS in ClinVar. We cannot be certain that the ultrahypermutator phenotype was caused by PMS2, POLE or a synergistic effect. The fact that the patient has such a high TMB could be consistent with this POLE variant driving the hypermutator phenotype. Also, the POLE mutation is associated with high proportions of C>A, C>T, and T>G variants [18], and a large number of these are observed in this patient's case.
Campbell noted that “Tumor types that show enrichment for MSI-MSI-H tumors cluster in the 10–100 Mut/Mb range, while tumors with mismatch repair and polymerase proofreading in the same types are ultrahypermutant.” The POLE variant may play a role in the mutator phenotype seen here.
6 The mention of the number of pages (supposedly of the genetic testing report?) is unnecessary and it would be preferable to give the exact number of somatic variants identified in both tumors for all categories indicated in the manuscript (instead of the broad statement “Both tumors had a very large number of somatic variants”). In contrast to what the authors mention (“The variants of uncertain significance were too numerous to count”), it is very important to count all somatic variants present in a tumor, in order to give an accurate number for the TMB, as well as to perform further genomic analysis. In fact, additional analyses (e.g., second hit testing, mutational signatures analysis) could be performed by using the germline-somatic next generation sequencing data generated to verify some of the authors’ hypothesis (maybe following some recommendation in Walsh et al. 2018 https://doi.org/10.1002/humu.23640). However, they would be out of the scope of the current case report.
6 Reply: You are absolutely right about this-thanks for encouraging a full count of VUS for each tumor. The ovarian tumor had 506 VUSs and the endometrial tumor had 657 VUSs. This was added to the paper. From the clinical side, we do not have access to the mutational sequence analysis at this time.
7 An appropriate comparison between the somatic mutations found in both tumors would also be helpful to clarify the similarities and differences between them, as well as to more clearly infer the alteration that came first. In this regard, the authors claimed in the last sentence of the manuscript that both carcinomas arose from one common site (endometrium) without further discussion or explanation. The authors merely indicated that there are 13 shared variants among one of the classes reported, but the lack of mention about percentages, other variant classes or variant allele frequencies, makes it difficult to assess the relevance of this finding.
7 Reply: Thank you-more detail was added to the paper with regard to percentage of variants in common. The ovarian tumor had 26 potentially actionable and biologically relevant variants of which 16 (61.5%) were in common with the endometrial tumor. The endometrial tumor had 34 potentially actionable and biologically relevant variants of which 16 (47%) were in common with the ovarian tumor. With the information we have available, we can only hypothesize which tumor arose first based on other authors and the clinical intuition of our Gynecologic Oncologist (JB) who treated this patient.
MINOR COMMENTS:
1 There are many issues regarding format, including for example figure legends showed before the figure (and not after it), errors in the references (ref. 3 - “In:”, ref. 10 & 13 – inconsistent doi, ref. 14 – inconsistent page number, etc.), abbreviations explained more than once (e.g., MMR), extra spaces in some sentences (e.g., line 160), numbers for the two tumors without clarification of which one is corresponding to each (e.g., TMB – line 120) an inconsistent reference within the text (ref. 2 – line 32) and missing methods section as mentioned above. The authors should carefully revise the format of their manuscript before resubmission.
1 Reply: Thank you for your careful review. Figure legends were adjusted to recommended format. We used Mendeley software to import citations and we have corrected those highlighted and verified all doi as active. Abbreviations were reviewed to eliminate duplications. Reference inconsistency was corrected.
2 The authors should avoid the terms “full/comprehensive tumor analysis/profiling” since they performed gene panel sequencing analysis of both tumors of the patient, which covered less than 1,000 genes and therefore less than 1% of the whole genome of the tumors. Although they are implementing in the clinic a technique that substantially increases the number of genes tested in comparison with previous methods, a comprehensive tumor characterization would require a whole genome sequencing analysis.
2 Reply: Thank you—eliminated the word comprehensive and full. While this testing is very thorough from the standpoint of what is available clinically right now, there are certainly much more comprehensive tests that can be done on a research basis, including whole exome/genome analysis. We appreciate you putting this in perspective.
3 Germline and somatic mutations identified should be compared/reviewed/assessed using publicly available databases (OMIM, ClinVar, gnomAD, COSMIC, etc.) apart from the ones used by the genetic testing companies performing the analyses.
3 Reply: Thank you. We did go back to the germline lab and request review for germline status, coverage and classification of the somatic mutations, most notably BRCA1/2. The PMS2 mutation was verified as likely pathogenic on ClinVar and that was added to the paper. We recognize that some variants on the somatic test may have varying classifications in the germline space and you gave a good example of that in the POLE.
4 The third paragraph of the results section should come before the second paragraph since it is about the germline testing mentioned in the first paragraph of the section.
4 Reply: Done, this makes more sense.
5 There are no mentions of figures 2 or 3 in the text of the manuscript. Done
Reference 1 should be changed for a more up-to-date study.
5 Reply: Added American Cancer Society Data from 2020. We chose the Pearlman paper because it was done in the Ohio state USA region of which we are a part, so we feel the data from this paper, while it is 4 years old is relevant to our clinical population. If it’s OK, we’d like to include it as well.
American Cancer Society. Cancer Facts and Figures 2021. https://www.cancer.org/research/cancer-facts-statistics/all-cancer-facts-figures/cancer-facts-figures-2021.html (accessed on 3 April 2021)
6 References are missing for the ovarian cancer epidemiology (line 27) and Lynch syndrome (line 28) statements in the introduction.
6 Reply: Reply: Thank you, added
7 Reference for Ambry Genetics statement in lines 170 and 171 is missing.
7 Reply: Added Vaughn et al
Vaughn CP, Hart KJ, Samowitz WS, Swensen JJ. Avoidance of pseudogene interference in the detection of 3' deletions in PMS2. Hum Mutat. 2011 Sep;32(9):1063-71. doi: 10.1002/humu.21540. PMID: 21618646.
8 CT and EGD abbreviations should be explained before use.
8Reply: Done
9 The resolution of all figures should be improved.
9 Reply: Tempus lab has provided the highest resolution figures possible.
10 Typos: *Immunohistorychemistry (line 37), *this (line 159). Inconsistency: extra-colonic – extracolonic (line 164).
10 Reply : Corrected

Reviewer 2 Report
The authors aim to report on a case with both endometrial and ovarian cancer in whom a PMS2 variant was identified. I am somewhat struggling to identify the specific aim or main message of this report. I also have some more specific comments:
Line 28: EPCAM itself is not a mismatch repair gene, when deleted it causes disregulation of MSH2
Line 40: Tumors with solitary loss of PMS2 expression
Line 43: consider reversing ovarian and endometrial, so the abbreviation makes more sense
Line 52: insert 'staining' for clarity
Line 68: please add the histological subtype of the polyp, does 'benign' mean hyperplastic?
Line 75: The PMS2 variant is not reported in the LOVD of PMS2 variants, so not curated by the InsiGHT. In ClinVar the variant is classified as likely pathogenic (class 4). Please add a statement on the pathogenicity of the variant.
Line 75: The PALB2 VUS is interesting in light of the positive family history for breast cancer, was segregation analysis performed?
Line 122: Were multiple sections of the tumor stained? Was there homogeneous weak staining? How do the authors explain the disconcordant staining for the ovarian and uterine tumors (for both PD-L1 and PMS2) in respect to their statement that the endometrial tumors was most likely the primary tumor
Line 131: I think the number of pages of variants is an unclear statement, I would suggest adding the number of variants or just state 'too numerous' and leave it at that.
Line 134: Did the authors check the frequency of these variants in the cosmic database?
Line 159: This sentence is confusing?
Line164: I do not agree with this statement that screening for extracolonic tumors is important based on the 60%, multiple papers reported no statistically/clinically relevant increase in risk for cancers other than colorectal and endometrial cancer for PMS2 carriers (Moller et al, Gut, 2018, ten Broeke et al, JCO, 2018)
Line 172: Was there a second hit in PMS2 identified? somatic variant or LOH? Did the authors look for muational signature 6?The authors mention a POLE variant in figure 5, this also causes a hypermutated phenotype. In other words, are the authors sure this tumor phenotype is caused by PMS2 loss?
Author Response
REVIEWER 2 Author's Reply
1 The authors aim to report on a case with both endometrial and ovarian cancer in whom a PMS2 variant was identified. I am somewhat struggling to identify the specific aim or main message of this report. I also have some more specific comments:
1 reply: The specific aim of this report is to present tumor profiling of a rare patient with SEOC in order to better understand the characteristics of tumors presumedly caused by germline PMS2. Our work also led to discovery of clonality of the two tumors.
2 Line 28: EPCAM itself is not a mismatch repair gene, when deleted it causes disregulation of MSH2
2 reply: Great point; this was corrected.
3 Line 40: Tumors with solitary loss of PMS2 expression
3 reply: Thank you for clarifying. This was changed as you directed.
4: Line 43: consider reversing ovarian and endometrial, so the abbreviation makes more sense
4 reply: Done, thank you.
5 Line 52: insert 'staining' for clarity
5 reply: Done, thank you.
6 Line 68: please add the histological subtype of the polyp, does 'benign' mean hyperplastic?
6 reply: Thank you, the full pathology detail was added for colonoscopy and EGD.
7 Line 75: The PMS2 variant is not reported in the LOVD of PMS2 variants, so not curated by the InsiGHT. In ClinVar the variant is classified as likely pathogenic (class 4). Please add a statement on the pathogenicity of the variant.
7 reply: Thank you. Some detail as to classification was added from the germline lab report and also from ClinVar.
8 Line 75: The PALB2 VUS is interesting in light of the positive family history for breast cancer, was segregation analysis performed?
8 reply: We were unable to perform segregation analysis. A family study was considered but only one affected relative was living and unavailable.
9 Line 122: Were multiple sections of the tumor stained? Was there homogeneous weak staining? How do the authors explain the discordant staining for the ovarian and uterine tumors (for both PD-L1 and PMS2) in respect to their statement that the endometrial tumors was most likely the primary tumor?
9 reply: Additional information was added in the methods regarding the PD-L1 staining.
10 Line 131: I think the number of pages of variants is an unclear statement, I would suggest adding the number of variants or just state 'too numerous' and leave it at that.
10 reply: Thank you- At your suggestion, we counted the variants rather than describe by pages and changed this in the paper.
11 Line 134: Did the authors check the frequency of these variants in the cosmic database?
11 reply: Yes, the Tempus xT clinical validation COSMIC also serves as an additional source of gene category claims.
https://www.oncotarget.com/article/26797/text/
12 Line 159: This sentence is confusing?
12 reply: Thank you. Adjusted sentence to clarify which variant gave which therapy indication.
13 Line164: I do not agree with this statement that screening for extracolonic tumors is important based on the 60%, multiple papers reported no statistically/clinically relevant increase in risk for cancers other than colorectal and endometrial cancer for PMS2 carriers (Moller et al, Gut, 2018, ten Broeke et al, JCO, 2018)
13 reply: We are in agreement that Moller et al states: “Heterozygous PMS2 mutation carriers were at small increased risk for colorectal and endometrial cancer but not for any other Lynch syndrome-associated cancer. “
In addition, NCCN Guidelines does not cite a high risk for extracolonic tumors outside of endometrial cancer. However, the Latham et al paper, while smaller, is more recent in 2020 and states: “While PMS2-related LS may have a more modest clinical phenotype, in this single-institution study, 60% (12/20) of patients with MMRD/MSI tumors presented with extra-colonic cancers. “ In addition, this patient does have a family history that includes 3 breast cancers and 1 pancreatic cancer on the paternal side, which is known to be the source of the mutation.
14 Line 172: Was there a second hit in PMS2 identified? somatic variant or LOH? Did the authors look for mutational signature 6? The authors mention a POLE variant in figure 5, this also causes a hypermutated phenotype. In other words, are the authors sure this tumor phenotype is caused by PMS2 loss?
14 reply: There was no recognized 2nd hit in PMS2 identified. We have added additional detail regarding POLE:
This very high TMB is considered as an ultrahypermutator tumor by Campbell et al [15]. It was noted that a POLE variant was identified on tumor profile testing in the endometrial tumor (POLE c.1306C>T p.P436S) and that polymerase proofreading alterations in POLE can play a role in ultrahypermutator phenotype. The POLE c.1306C>T p.P436S variant was reviewed by Tempus Lab variant scientist and functional evidence led to a somatic pathogenic classification. The evidence used to classify POLE c.1306C>T p.P436S as pathogenic includes:
1. Variant causes increased mutagenesis in yeast assays [16]; variant is referred to as p.P451S in yeast).
2. Variant identified in a patient whose phenotype is similar to patients with known POLE mutations[17].
3. Variant falls in the POLE exonuclease domain (codons 269-485). While this evidence is sufficient to classify the variant as pathogenic in a somatic context, it does not quite meet the threshold for being classified pathogenic as a germline variant and is seen as germline VUS in ClinVar. We cannot be certain that the ultrahypermutator phenotype was caused by PMS2, POLE or a synergistic effect. The fact that the patient has such a high TMB could be consistent with this POLE variant driving the hypermutator phenotype. Also, the POLE mutation is associated with high proportions of C>A, C>T, and T>G variants [18], and a large number of these are observed in this patient's case.

Reviewer 3 Report
the present paper describes a case of a 39-year-old SEOC patient with germline Lynch syndrome with a complete clinical tumor analysis and accurate molecular characterization regarding the origin of these tumors with potential therapy options. Next generation sequencing for 648 gene identified sixteen actionable and biologically relevant shared somatic mutations. The work is interesting and well written. I recommend more accurate and higher magnification histologic figures to highlight the immunohistochemical detail of both tumors. Also, the magnification should be indicated in the caption. I also suggest including additional recent literature related to Lynch syndrome, particularly in consideration of prognosis. Three papers have recently been published, two of them in MDPI journals.
- Stelloo E, Smit VTHBM, et al. Prevalence and Prognosis of Lynch Syndrome and Sporadic Mismatch Repair Deficiency in Endometrial Cancer [published online ahead of print, 2021 Mar 6]. J Natl Cancer Inst. 2021;djab029. doi:10.1093/jnci/djab029
- Dondi G, Coluccelli S, De Leo A, et al. An Analysis of Clinical, Surgical, Pathological and Molecular Characteristics of Endometrial Cancer According to Mismatch Repair Status. A Multidisciplinary Approach. Int J Mol Sci. 2020;21(19):7188. Published 2020 Sep 29. doi:10.3390/ijms21197188
- Deshpande M, Romanski PA, Rosenwaks Z, Gerhardt J. Gynecological Cancers Caused by Deficient Mismatch Repair and Microsatellite Instability. Cancers (Basel). 2020;12(11):3319. Published 2020 Nov 10. doi:10.3390/cancers12113319
Author Response
Thank you for your time in review and the additional references that enrich this submission. Tempus lab has added IHC figures. In addition, we added significant methodology at the request of other reviewers. Please see attachments with changes.

Round 2
Reviewer 1 Report
I am sincerely grateful for the thorough review of the manuscript performed by the authors, especially regarding the addition of the methods, which was my main point in the previous report. I would suggest for the future always including a new version of the manuscript highlighting the added/modified text with the resubmission. This makes peer-reviewing much easier. Besides, I have some minor comments that you can find below. Other than that, I am
delighted to help in building a definitely much stronger manuscript. Great job.
Minor comments
1. Please provide a reference for the following statement on line 206: “Functional studies demonstrate that S417Y leads to a partial reduction of ChAM-mediated PALB2 chromatin association, without affecting the cellular resistance to CPT”.
2. I would again suggest removing the reference to any percentile (lines 236, 307 and 309) regarding TMB since it makes no sense without a larger cohort to compare and you already reported the number of mutations per megabase, which is the standard unit used in the field.
3. Please add a mention to the mutational signatures caused by mutations in the exonuclease domain of POLE (mutational signatures SBS10a, SBS10b and SBS28:
https://cancer.sanger.ac.uk/signatures/; Alexandrov et al. 2020 Nature:
https://doi.org/10.1038/s41586-020-1943-3) on line 250, apart from mentioning its predominant types of single base substitutions.
4. As previously mentioned, proper numerical comparison of the VUS between the ovarian and the endometrial tumor should be included in the manuscript instead of thestatement on line 319: “The VUSs were too numerous to compare directly”.
Author Response
- Please provide a reference for the following statement on line 206: “Functional studies
demonstrate that S417Y leads to a partial reduction of ChAM-mediated PALB2
chromatin association, without affecting the cellular resistance to CPT”.
Thank you. Ambry lab used data from this reference in determining their classification of the PALB2 variant. We added the Bleuyard paper as reference 15.
Bleuyard JY, Butler RM and Esashi F. Perturbation of PALB2 function by the T413S mutation found in small cell lung cancer [version 2; peer review: 3 approved]. Wellcome Open Res 2018, 2:110 https://doi.org/10.12688/wellcomeopenres.13113.2
- I would again suggest removing the reference to any percentile (lines 236, 307 and 309) regarding TMB since it makes no sense without a larger cohort to compare and you already reported the number of mutations per mega base, which is the standard unit used in the field.
Thank you for pointing out that the mutations per mega base is a more standard unit. We were getting too focused on the percentile designated by one clinical lab. The percentile reference was removed.
- Please add a mention to the mutational signatures caused by mutations in the
exonuclease domain of POLE (mutational signatures SBS10a, SBS10b and SBS28:
https://cancer.sanger.ac.uk/signatures/; Alexandrov et al. 2020 Nature:
https://doi.org/10.1038/s41586-020-1943-3) on line 250, apart from mentioning its
predominant types of single base substitutions.
Thank you. We appreciate you strengthening the scientific background in this paper as we are coming from the clinical side. The COMIC and Alexandrov references were added and line 251 now reads:
Alexandov has described mutational signatures caused by mutations in the exonuclease domain of POLE, including single base substitutions: SBS10a, SBS10b and SBS28 referenced in the Catalogue of Somatic Mutations In Cancer. Tumors with these mutational profiles generate a large volume of somatic mutations (>100 mut/MB) and are termed as hypermutators [20, 21].
- As previously mentioned, proper numerical comparison of the VUS between the ovarian and the endometrial tumor should be included in the manuscript instead of the
statement on line 319: “The VUSs were too numerous to compare directly”.
Thank you. We changed the wording at line 334 to:
In addition to the potentially actionable and biologically relevant variants, number of VUSs was also reported for each tumor. The ovarian tumor had 506 VUSs and the endometrial tumor had 657 VUSs. We were unable to compare the VUSs directly for commonality.
Reviewer 2 Report
Thank you for the clear responses and making the suggested changes. Just two small remarks/suggestions:
- Regarding the statement about 60% showing non-colonic, non-endometrial cancer. I would still like to see a statement here also citing other work and not just Latham et al, as the authors said, this is a small PMS2 cohort. In my opinion it is important to not just cite one paper supporting the authors conclusion. Also the fact that breast cancer occurs in the family where the mutation originates does not mean these tumors are PMS2-associated.
- Point 14: I think it would be informative for readers to mention that no second hit in PMS2 was identified.
Author Response
Thank you for helping us make this a better submission. Our reply is below in italics. We have added your suggestions.
Point 1: Regarding the statement about 60% showing non-colonic, non-endometrial cancer. I would still like to see a statement here also citing other work and not just Latham et al, as the authors said, this is a small PMS2 cohort. In my opinion it is important to not just cite one paper supporting the authors conclusion. Also the fact that breast cancer occurs in the family where the mutation originates does not mean these tumors are PMS2-associated.
We understand your point about focusing on the primary risks with PMS2 rather than the exceptions with small sample size. We added the more comprehensive Lynch Database reference by Moller and adjusted the paragraph at line 293 to read:
"This patient had a personal and family history of cancer that is not consistent with the moderate-risk PMS2 Lynch syndrome risk profile in NCCN [3]. The Prospective Lynch Syndrome Database has reported primarily colorectal and endometrial cancer risk associated with PMS2 [19]. This patient's personal and family history included ovarian cancer, three cases of breast cancer as well as pancreatic cancer. These are all at lower risk in PMS2 germline carriers, but breast cancers have been reported in some PMS2 Lynch patients [20]. Although PMS2 mutation carriers do have lower overall cancer risk compared to other Lynch mutations carriers, a smaller study of PMS2 LS patients showed that 60% of patients with MMR deficient/MSI tumors presented with extra-colonic cancers [21]. NCCN Guidelines primarily address surveillance and prevention strategies for colon and gynecologic cancers with PMS2 germline mutation. Clinicians may consider family history to determine any indication for screening other cancers."
Point 14: I think it would be informative for readers to mention that no second hit in PMS2 was identified.
At line 237, a statement was added on the PMS2 2nd hit as suggested. This reads:
"There was no recognized 2nd hit, or mutation, in PMS2 identified."
Tracked changes are attached. Thank you for your time and persistence in helping us get this right.
